# Bond Behaviour of Near-Surface Mounted Strips in RC Beams—Experimental Investigation and Numerical Simulations

**DOI:** 10.3390/ma14164362

**Published:** 2021-08-04

**Authors:** Renata Kotynia, Hussien Abdel Baky, Kenneth W. Neale

**Affiliations:** 1Department of Concrete Structures, Lodz University of Technology, Al. Politechniki 6, 90-924 Lodz, Poland; 2The Shockey Precast Group, Winchester, VA 22603, USA; h_abdelbaky@hotmail.com; 3Department of Civil Engineering, University of Sherbrooke, Sherbrooke, Quebec, QC J1K 2R1, Canada; Kenneth.Neale@USherbrooke.ca

**Keywords:** carbon fibre reinforced polymers (CFRPs), modified reinforced concrete beams, flexural strengthening, debonding, NSM, 3-D finite element analysis

## Abstract

This paper presents an investigation of the bond mechanism between carbon fibre reinforced polymer (CFRP) laminates, concrete and steel in the near-surface mounted (NSM) CFRP-strengthened reinforced concrete (RC) beam-bond tests. The experimental program consisting of thirty modified concrete beams flexurally strengthened with NSM CFRP strips was published in. The effects of five parameters and their interactions on the ultimate load carrying capacities and the associated bond mechanisms of the beams are investigated in this paper with consideration of the following investigated parameters: beam span, beam depth, longitudinal tensile steel reinforcement ratio, the bond length of the CFRP strips and compressive concrete strength. The longitudinal steel reinforcement was cut at the beam mid-span in four beams to investigate a better assessment of the influence of the steel reinforcement ratio on the bond behaviour of CFRP to concrete bond behaviour. The numerical analysis implemented in this paper is based on a nonlinear micromechanical finite element model (FEM) that was used for investigation of the flexural behaviour of NSM CFRP-strengthened members. The 3D model based on advanced CFRP to concrete bond responses was introduced to modelling of tested specimens. The FEM procedure presents the orthotropic behaviour of the CFRP strips and the bond response between the CFRP and concrete. Comparison of the experimental and numerical results revealed an excellent agreement that confirms the suitability of the proposed FE model.

## 1. Introduction

In the last two decades, an enormous amount of research work and numerous innovative applications have been published in the field of externally bonded (EB) fibre-reinforced polymer (FRP) concrete structures. The near-surface mounted technology has become very effective for flexural strengthening in single-span and double-span members, especially in the negative moment regions, where externally bonded FRP reinforcement is sensitive to environmental and mechanical impacts. Due to much better anchorage, the near-surface mounted (NSM) reinforcement is more safe and efficient than EB FRP applications [1,2,3,4,5,6,7,8,9,10].

The key factor affecting the performance of CFRP-strengthened structures is the bond phenomenon between the CFRP bars or strips and concrete. Although the bond mechanisms have been extensively studied over the last few years [11,12,13,14,15,16,17,18,19], debonding mechanisms of the NSM CFRP strips depend on the interfacial/flexural crack width along the bond length. The interactions between the internal reinforcement and the externally bonded NSM strips on the stress transfer between the CFRP laminates and surrounding concrete have been published in [20,21,22,23,24,25,26,27,28].

The flexural behaviour of CFRP-strengthened concrete beams and slabs has been deeply investigated in RC members, which confirmed significantly higher efficiency compared to the EB configuration [29,30,31,32,33,34,35,36,37]. In fact, the reason for this benefit is much better CFRP to the concrete anchorage of CFRP reinforcement in adjacent concrete cover than the EB technique based on the surface application inherently related to the bond behaviour between the CFRP and concrete. Debonding of CFRP laminates implies that a full understanding of the bond mechanism between the EB reinforcement and concrete is the crucial problem. Many shear tests (single shear, double shear and beam-bond test) have been widely used in experimental studies to precisely investigate the bond mechanism between the CFRP and concrete [31,32,33,34,35,36,37,38,39,40,41,42,43,44]. However, with few exceptions, all previous bond tests on NSM CFRP strip or bar concrete interfaces were performed on the direct shear or modified beam specimens conducted on plain concrete, without taking into account the influence of internal steel bars on the bond mechanisms between the CFRP and concrete. This is because the beam-bond tests are much more complicated and accounting for the effect of internal steel bars and the NSM CFRP reinforcement on the bond mechanics with all associated parameters hindered the analysis and the understanding of the failure mechanisms.

This is why the modified beam-bond tests according to the RILEM [45] have been implemented in only a few tests to investigate the effect of the flexural behaviour with crack propagation and beam curvature on the CFRP to concrete bond performance [46,47,48,49].

The bond mechanisms of CFRP to concrete joints depend on the interfacial crack widths along with the interface, which means that the steel reinforcement ratio can significantly affect the bond performance between the CFRP and concrete [49,50,51,52,53,54,55,56,57,58,59,60,61,62]. Furthermore, comparing the results of various bond tests often leads to discrepancies, especially concerning the influence of the compressive concrete strength on the bond mechanisms when applying the NSM technique. An extensive description of the NSM tests presented in [63] confirmed that NSM FRP strengthened concrete members tested under monotonic and fatigue loads confirmed much better performance and serviceability than the EB FRP technique. Fatigue tests of RC beams strengthened with NSM CFRP strips indicated more extended fatigue resistance than EB CFRP laminates. The utilization of graphene oxide within the innovative cementitious adhesive (IHSSC-CA) showed outstanding properties and indicated that the existing polymer-based cementitious adhesive increases the NSM CFRP bond strength, stiffness, fatigue life, resistance to high temperature and serviceability limit state [63].

The interesting test results of the NSM technique extended with Steel Reinforced Grout (SRG) materials published in [64] indicated many advantages: high strength to weight ratio, easy application, high durability and low costs of strengthening.

The effectiveness of the strengthening technique using either externally bonded reinforcement (EBR) or NSM procedures was found to be dependent on various parameters, such as the steel reinforcement ratio, distribution of the bending moments along the beam span, a number of layers of the externally bonded CFRP laminates, their axial stiffness and the bond length.

As far as related theoretical work is concerned, to achieve an accurate prediction for the bond strength, one must first understand the shear stress transfer mechanisms along the interface, and also determine the multiaxial state of stress to which a point inside the interface is subjected.

The finite element simulations based on a fairly sophisticated micromechanics material description, known as the microplane theory, were used to model the behaviour of tested specimens. Adopting such a concrete theory has been taken foremost because this approach leads to great success in representing the concrete behaviour under the complex stress–strain state. The flexural failure modes and the associated concrete cover splitting can be simulated using a major discrete crack approach along with the microplane concrete models [65,66,67,68].

In the finite element model proposed in this study, an accurate bond-slip relationship is used to represent the interfacial behaviour of CFRP to concrete interfaces. The relations were developed by [18] and account for the state of stress in the concrete adjacent to the adhesive along the NSM and EB CFRP strips and bars.

The purpose of the research is the comparative analysis of the numerical and experimental results to show the compatibility of the presented model in reference to obtained test results of the NSM CFRP-strengthened concrete members.

## 2. Experimental Program

### 2.1. Materials

Two different concrete batches with the concrete mix proportions for cylinder compressive strength 20 MPa and 40 MPa, summarized in Table 1, were used in the experimental program published in [48]. The average cylinder compressive strength, $f_{c}^{\prime }$ and the tensile splitting strength, $f_{t}$ were determined for each beam, and they are summarized in Table 2 (columns 5 and 6, respectively). Both cylinder samples (150 mm in diameter and 300 mm in height) and prism specimens (150 × 150 × 150 mm) were prepared during casting of each concrete batches. It should be mentioned that the nomenclature of several beams has been changed according to the original publication [48].

The beams were reinforced in the tensile zone with longitudinal steel ribbed bars of nominal diameters: 8 mm or 16 mm, and steel stirrups of a nominal diameter of 8 mm at a spacing of 100 mm [48]. The reinforcement in the compressive zone of all tested beams was made of steel ribbed bars of a nominal diameter of 8 mm. The experimental modulus of elasticity of the steel reinforcement, $E_{s}$, its tensile strength, $f_{su}$, and yield stress, $f_{ys}$, are summarized in Table 3.

CFRP strips (Type XS1.524) of 2.41 mm thickness and 15 mm width were used for the NSM strengthening of the RC specimens and their mechanical characteristics are presented in Table 4. The strength characteristics of the CFRPs (ultimate tensile strength, $f_{fu}$, elastic modulus in the fibre direction, $E_{f}$, and ultimate strain, $\varepsilon_{fu}$) are summarized in Table 4. The Sikadur^®^30 epoxy adhesive was used to bond the CFRP strips to the concrete.

### 2.2. Test Specimens

The experimental tests were carried out on four main series consisting of thirty single-span, simply supported, modified RC beams published in [48]. The RILEM configuration (1982) [45], originally devised for bond tests of steel reinforcement to concrete, was adapted to the current CFRP to concrete beam-bond tests. The test setup consisted of two separate concrete blocks connected with a steel hinge at the beam mid-span in the compression zone and the internal longitudinal steel reinforcement on the tension side. The key advantage of this setup was to investigate the bond performance of the CFRP to concrete interface under the beam curvature without being influenced by the complexity of the major flexure beam cracks in the mid-span. Another reason for using this test setup was to allow the forces to transfer from the NSM CFRP strips to the concrete blocks in a similar way to that of direct shear tests rather than from the concrete to the NSM CFRP strips as is the case of ordinary beam tests.

The test specimens described in [48] had two different rectangular cross sections, two beam spans, and two steel reinforcement ratios. Two cross sections of 150 × 200 mm and 150 × 400 mm, referred to as Series “I” and “II”, respectively. Two beam spans of 1350 mm and 2000 mm, are denoted by symbols “S” (short) and “L” (long), respectively. Two internal steel reinforcements contained 2 #8 and 2 #16, corresponding to the notations: “A” and “B”, respectively, implemented for both beam cross sections and both beam spans. Two concrete compressive strengths of 20 MPa and 40 MPa were used in the experimental program.

Twenty-one beams were strengthened with a single CFRP strip with the bond length ranging from 80 mm to 160 mm. Three beams were strengthened with double strips at a spacing of 50 mm and the bond length of 80 mm (beams: NILA/50/2 × 80, NIISB/40/2 × 80, NIILB/40/2 × 80). The objective of these specimens was to address the effect of double NSM CFRP strip areas comparing to the single CFRP strip on the flexure capacity of the tested specimens. The last six beams in the experimental program were tested as unstrengthened (NISA/30, NISA/20, NISB/20, NILA/40, NILA/50, NILB/40).

The first number in the specimen ID (column 4, Table 2) refers to the designed concrete compressive strength. The real measured concrete strengths are listed in (column 5, Table 2). The last number in the specimen ID describes the bond length measured from 125 mm away from the mid-span (100 mm from the internal edge of the beam), as is shown in Figure 1.

Two beams (NILB/40/120s, NILA/40/120s) had the internal steel bars cut at the mid-span to investigate the effect of a lack of continuity of the internal reinforcement on the CFRP bond strength. Cutting the internal steel bars at the mid-span was designed to simulate the case of the direct shear test where the forces transferred from the CFRP strips to the concrete without any interaction of the internal steel bars. However, as the result of the fact that the internal reinforcement remained inside the concrete beams, it was expected to affect the flexural cracks at the NSM CFRP cut-off point. The objective of testing these two particular beams was to better understand the role of the internal steel bars on the bond performance.

### 2.3. Instrumentation and Test Set-Up Description

The test specimens were cast in a horizontal steel mould. The reinforcing skeletons for the beams were prepared in the laboratory of the Concrete Structures (TUL) [48]. Preparation of each member started with the mounting of two rectangular prisms made of polystyrene foam in the centre of each skeleton (70 × 90 × 150 mm and 125 × 50 × 150 mm in beams type “I”, and 75 × 90 × 150 mm and 325 × 50 ×150 mm in beams type “II”) to provide a space for the steel joint placed in the compressive part of the beam and a space in the full depth of the beam presented in Figure 2. After beam casting, the foams were removed.

All specimens were tested in an upside-down position to observe cracking patterns of the beams and to monitor the strain readings of the CFRP strips. The beams were tested in a four-point bending static scheme using hydraulic servomotors, as is shown in Figure 3.

Axial concrete strains in the tensile (R1–R5) and compressive (R6–R10) zones were monitored with the linear variable displacement transducers (LVDTs; type PSx10). The LVDT measurements were transferred to strain readings using the special Data Acquisition System (DAS) collecting the test data. The compressive and tensile strain measurements were registered in the level of the longitudinal steel hinge axis and in the level of the longitudinal steel reinforcement, respectively. The locations of all gauges are shown in Figure 3a,b. The strains in the NSM CFRP strips were recorded using 5 mm long electric strain gauges bonded in the symmetry axis of the CFRP width (Figure 3c). The slip between the NSM strips and the adjacent concrete in some beams was registered with LVDT gauges (Figure 3c). The measurements were automatically recorded at each loading level using DAS connected with the computer.

## 3. Test Results

The experimental results are presented with respect to the failure modes, ultimate loads and strain measurements published in detail in [48].

### 3.1. Failure Modes

Failure modes of the tested NSM CFRP strengthened beams occurred by the strip debonding together with the surrounding concrete cover along or slightly above the level of the steel reinforcement [48]. In the beams with longitudinal steel reinforcement, the CFRP debonding occurred in three slightly different planes of concrete, characterized by flexural cracks along the beam span. The first crack initiated at the strip’s end and then followed along with the internal steel reinforcement through the concrete interface, resulting in a cone shape (Figure 4a,b), or cracking observed on the entire width of the beam (Figure 4c,d). However, the beams with the cut steel reinforcement failure mode were different from those with longitudinal reinforcement, characterized by the CFRP debonding from the surrounding concrete cover, above the level of the internal steel reinforcement (Figure 4e).

The specimens with cut steel bars at the mid-span indicated the CFRP strip’s debonding between the CFRP end and the adhesive layer. In several beams, a significant slip of 5 mm between the CFRP strip and concrete occurred (Figure 4e—Point “1” moved at the failure to Point “2” and Point “1′” became Point “2′” after debonding).

The CFRP debonding with a cone shape (Figure 4a,b, case of small bond length) has been commonly observed in pull–out tests of NSM CFRP in concrete prisms, while the CFRP debonding on the entire width of the beam (Figure 4c,d, case of long bond length) has been more relevant for NSM CFRP-strengthened beams, labelled as plate-end debonding or concrete cover splitting (Figure 4c,d).

For the unstrengthened beams, a typical failure mode for conventional reinforced concrete beams was observed due to the steel yielding that generally follows by steel bar rupture. For the specimens with the long CFRP bond length, the failure load was slightly higher than the yield load of the internal steel bars at the cross section without NSM strips (cut-off point of NSM CFRP strips). Once the internal reinforcement yielded at this specific location, the major flexural crack at the cut-off point of the NSM CFRP strips merged with the horizontal crack that was initiated at the steel-concrete interface. This indicated that, for a longer bond length, the failure mode began as flexural failure due to steel yielding followed by bond failure in the plane located between the bottom of the CFRP strips and the upper surface of the internal steel bars, or at the level of the steel–concrete interface. On the other hand, for the shorter bond length, the failure mode was typically bond failure at the adhesive and concrete interface.

### 3.2. Crack Patterns

The observed crack patterns for all tested specimens were similar to those reported in the literature for the CFRP—strengthened beams. The crack concentration area was located in the bonded CFRP region; however, cracks first initiated at unbonded locations. The first crack appeared at the cross section of the CFRP cut–off with the shorter bond length at the beam’s mid-span. Then, the flexural cracks followed at the beginning of the CFRP laminates on the other beam side (crack No. 1 in Figure 5a). This resulted from the sudden reduction in the flexure capacity of the beam cross-section at the end of the NSM CFRP strips. Other flexural cracks occurred within the beam out of the CFRP reinforcement (cracks No. 2 and 3 in Figure 5a). Just before the failure of the beam, more cracks appeared along the bonded CFRP strips on both sides of the beam (Figure 5b,c).

Horizontal cracks at the level of the longitudinal reinforcement were observed at a load level preceding the CFRP debonding for the beams strengthened with double NSM strips, having the larger steel reinforcement (2 #16) (compare Figure 5a,b). This load level was slightly higher than the yielding load level of the internal reinforcement. The cracks were initiated mostly due to the bond failure at the steel–concrete interface.

### 3.3. Ultimate Loads

The experimental ultimate loads for the strengthened and unstrengthened beams ($f_{u}$, $f_{u0}$) and corresponding CFRP strains measured at the bottom strips at the mid-span, $\varepsilon_{fb}$, are summarized in Table 2. The strengthening efficiency is defined by the ratio $f_{u}/f_{u0}$. Test results presented in Table 2 indicate that for the NSM CFRP strengthening with the lowest reinforcement ratio, the lowest strengthening enhancement ratio of 6% was observed for beams with compressive concrete strength of 20 MPa, referring to specimen NISA/20/130, while the highest strengthening efficiency of 24% occurred in the specimens with compressive concrete strength of 32.5 MPa (specimen NISA/30/80). Surprising was a decrease in the ultimate load of the beams reinforced with the higher steel reinforcement strengthened with NSM strips of 85 mm and 130 mm bond lengths (respectively, by 22% for beam NISB/20/85 and 28% for beam NISB/20/130). The reason for this decrease was the higher stiffness of the beam with the higher steel reinforcement compared to the beams with lower reinforcement that indicated higher ductility before and after strengthening.

The increase in the ultimate load of the longest span beams was observed in the range between 20% and 73%, depending on the compressive concrete strength and the bond length of the CFRP strip (NILA/40/100 with $f_{c}^{\prime }$ = 41.75 MPa and NILA/40/120 with $f_{c}^{\prime }$ = 38.5 MPa). The cutting of the internal reinforcement at the mid-span significantly delayed the CFRP debonding and decreased the ultimate load of the specimens by 452% for the beams with a higher reinforcement ratio (beams NILB/40/120 and NILB/40/120s), and only 82% for the beams with lower reinforcement ratio (beams NILA/40/120 and NILA/40/120s).

The experimental results of the specimens strengthened with NSM reinforcement (Table 2) indicated that the bond length of the CFRP strips did not significantly affect the enhancement of the ultimate load. The reason was the failure mechanism of all tested beams by concrete cover splitting. No significant slip was observed between the CFRP strips and adjacent concrete, besides the specimens with cut the tensile steel reinforcement. This indicated that the flexural capacity of the beams (due to steel yielding) controlled the ultimate load-carrying capacity of these specimens rather than debonding or slippage of the CFRP strips. As expected, for these cases the debonding strain levels in the CFRP strips appeared in the same range as the steel yielding strains. This observation highlights the influence of the internal steel reinforcement for the CFRP on concrete bond behaviour. The steel bars indicated changing of the concrete failure plane from a trapezoidal shape to a mainly horizontal shape, appearing along or slightly below the tensile steel reinforcement.

### 3.4. Debonding Strains and Strain Distributions

Table 2 shows the axial strains in the CFRP laminates, $\varepsilon_{fb}$, and the strains in the tensile internal reinforcement, $\varepsilon_{su}$, at the ultimate load level. The strain profiles in the CFRP strips measured from the beginning of the CFRP bond length are shown in Figure 6 for the half of each beam having the shortest bond length. The maximum strain in the CFRP strip was registered along the un-bonded length; however, the strain values decreased slightly up to the cut-off point of the CFRP strip. Afterwards, the strains in the CFRP decreased progressively along the bond length, which indicated that the tensile stress was transferred from the CFRP strip to the adjacent concrete.

Figure 6a–d shows the effect of cutting the internal steel bars on the CFRP strain profiles. This caused a delay of CFRP strip debonding; accordingly, the specimens with cut steel bars achieved higher strain values in the CFRP strips than those of the beams with continuous steel bars, with an average value of 58% (respectively, for beams NILA/40/120 and NILA/40/120s; Figure 6a,c). However, the ultimate load reduced from 24.30 kN to 13.33 kN, indicating that the highest strain values in strengthening CFRP strips can be obtained for beams without internal steel bars. A similar observation was found for the beams NIISB/40/120 and NIISB/40/120s, which indicated 75.6% higher CFRP strains after cutting steel reinforcement (Figure 6b,d). However, the ultimate load of the specimen NILB/40/120s (Figure 6d) was 18% of that of Specimen NILB/40/120.

## 4. Finite Element Modelling

A 3D displacement—controlled nonlinear micromechanics—based finite element analysis of the tested specimens was carried out. All tested beams were modelled and simulated numerically. The major discrete crack approach was adopted in this study to capture various failure modes. Similar simulations have been presented in the literature for FRP—strengthened beams using external laminates [59,69,70,71,72,73,74,75,76]. In this approach, interface elements were aligned normally to the anticipated splitting planes (i.e., planes at which the CFRP strip detached from the concrete beam as observed in the experimental tests), as will be shown later. The flexural failure modes and the associated concrete cover splitting were simulated using a major discrete crack approach, along with the microplane concrete model. In addition, the interface elements characterized by the nonlinear bond stress—slip law—were implemented to represent the relative displacement between the CFRP strip and concrete.

The debonding mechanism and the shear behaviour of the concrete were functions of crack propagation along with the FRP/concrete interface. Therefore, it was not accurate to represent the concrete shear behaviour using the constant shear modulus for concrete after cracking as in the case of the smeared crack approach. This is the main reason for implementing a microstructure-based constitutive law for the concrete. In the microplane appreciate, another technique was used to represent the concrete behaviour, including shear nonlinearity, the interaction between shear and compression or tension, and the crack propagation and opening. In the microplane model, the microscopic shear boundary represents friction. In that, a yield function for friction was used to simulate the shear and normal stress. Moreover, the microscopic tensile deviatoric boundary condition was adopted to represent the crack opening by simulating the volumetric expansion and lateral strains of unconfined compression tests.

The microplane constitutive law was implemented as an additional part of the software ADINA. The constitutive laws for the steel and CFRP coded in the software were employed in the analysis. Details of these constitutive laws are included in the ADINA theory and modelling guide [70] and they are further described in the below section.

### 4.1. Material Models for Concrete, Steel and CFRP

The constitutive theory shows the microstructural level of the material considering three-dimensional elements defined by a set of microplanes of different orientations arranged in a regular pattern. The general microplane model shown in Figure 7 describes in detail the material element, which is characterized by twenty-eight equally located planes per hemisphere. These planes show the damaged or weak planes at the microstructural level presented in the aggregate mortar interface and the planes of the microcrack formation. The orientation of the single plane is defined by a unit normal vector $n_{i}$. The main assumption of this theory is a macroscopic strain tensor, $\varepsilon_{ij}$, which is projected into a microscopic normal strain vector $\varepsilon_{n}$ and a shear strain vector $\varepsilon_{t}$ on a separate microplane.

The main approach presented in the microplane model is to project the macroscopic strain tensor into components on a certain number of microplanes in order to define each material element. Stress–strain function on the microplane is determined by the respective micro-stress components. The application of the principle of the virtual work and the numerical integration over all microplanes is used to determine the macroscopic stress components. The original M4 microplane model was published in [71,72]. It involves highly nonlinear stress–strain relations. In the microplane model M4, the values of the non-dimensional material constants are defined by the microplane model ($c_{1}$ to $c_{18}$), adopted from the main publication [71]. The non-dimensional tailored material parameters ($k_{1}$ to $k_{4}$), the macroscopic Young’s modulus, $E_{c}$, and the Poisson’s parameter, v, are determined separately for each concrete batch. As the confining stress is not enough to cause the pure concrete failure, the parameter defining the triaxial compressive behaviour under a high confining state of stress $k_{2}$, and the parameters describing the concrete behaviour under hydrostatic compression ($k_{3}$, $k_{4}$), were included for the specimens according to:(1)$$k_{2}=110.0;k_{3}=12.0;k_{4}=38.0;$$
the Poisson’s parameter, v equals 0.18 and the parameter $k_{1}$ is computed for each concrete batch according to the formula:(2)$$k_{1}=2.45\times 10^{-4}\frac{\;f_{c}^{\prime }}{1150E_{c}}$$

The concrete characteristics presented by elasticity modulus and compressive strength ($E_{c}$ and $f_{c}^{\prime }$) are taken from the experimental data for each concrete batch in GPa and MPa, respectively (Table 2).

The behaviour of the major cracks in the concrete (Figure 8) is modelled considering the major discrete crack approach that has been adopted by several researchers [65,67]. Interface elements were aligned to bridge the anticipated locations of the two main cracks. The first major crack was the flexural crack initiated at the cut-off point of the CFRP strip, propagating vertically. To account for the geometric discontinuity arising from concrete cover splitting, another major discrete crack was assumed along with the steel–concrete interface (Figure 8). The concrete tensile strength was used to constitute the material response of these elements in tension, with a large strength value in compression (Figure 8).

A bilinear elastic–plastic model was used to represent the behaviour of the steel reinforcement. The tangent modulus in the strain-hardening zone was assumed as 1% of the elastic modulus. For the CFRP composites, a linear elastic model was used to represent the behaviour up to rupture. The elastic modulus in the transverse direction, $E_{t}$, equals 10% of that in the longitudinal direction, $E_{f}$.

### 4.2. Interface Model between CFRP and Concrete

The bond tests of NSM FRP to concrete joints developed by [6,36,37,38] have been summarized by [38] and are presented in Figure 9 and Equation (3):(3)$$\tau \left(s\right)=\left\{\begin{array}{c} \tau_{m}{\left(\frac{s}{S_{m}}\right)}^{\alpha }s\leq S_{m}\\ \tau_{m}{\left(\frac{s}{S_{m}}\right)}^{-\alpha }s>S_{m} \end{array}\right.$$

The value $S_{m}$ was in the range of 0.1 to 0.3. The bond slip shown in Equation (3) considers the average bond stress $\tau_{m}$ instead of the maximum bond strength $\tau_{\max}$ (the index α was changed in Equation (3) for −α), in case a bond-slip relationship is proposed based on the mathematical derivation presented in the previous section $\tau_{\max}$ (Figure 8).

Experimental results by [40,41,52] indicated the rational initial slope depending on elastic responses of FRP, adhesive and concrete rather than an infinite slope of equation bond-slip curve characterized by Equation (3). Furthermore, the ascending and descending branches of the curve have to be smoothly connected (at $s=S_{o}$ $\tau =\tau_{\max}\;$ and $\frac{\partial \tau }{\partial s}=0$). At initial loading, the interface behaves elastically and the initial slope of the bond-slip model can be determined from the theory of elasticity. The parabolic function of the bond was used to describe the ascending part of the bond-slip curve shown in Figure 9 and calculated based on formulas:(4)$$\tau =\mathrm{bs}+{\mathrm{as}}^{2}\;s\leq S_{o}$$

The constants a and b in Equation (4) were obtained from the following boundary conditions:(5)$$s=0\;\frac{\partial \tau }{\partial s}=E_{o}$$
(6)$$s=S_{o}\;\;\tau =\tau_{\max},\;\frac{\partial \tau }{\partial s}=0$$
(7)$$\frac{1}{E_{o}}=\frac{t_{f}}{G_{f}}+\frac{t_{a}}{G_{a}}+\frac{h_{\mathrm{eff}}}{G_{c}}$$
where $h_{\mathrm{eff}}$ is the thickness of the interfacial layer inside the concrete that transfers stresses from FRP to the concrete block and $t_{f}$ is the depth of FRP strip or the bar diameter, $t_{a}$ is the thickness of the adhesive layer, $G_{f}$, $G_{a}\;$ and $G_{c}$ are the shear modulus of the FRP composites in transversal directions, adhesive and concrete, respectively.

Combining these boundary conditions with Equation (4), the shear stress, τ, the behaviour of the interface between the CFRP and the concrete beam is modelled as a function of the relative displacement, s, between the two sides of the interface (concrete and CFRP). This function, as proposed, was proposed in publications [74,75,76]. The τ−s relationship was controlled by equations:(8)$$\tau =\left\{\begin{array}{c} \tau_{\max}\left[\left(\frac{2s}{S_{o}}\right)-{\left(\frac{s}{S_{o}}\right)}^{2}\right]\;\;s\leq S_{o}\\ \tau_{\max}e^{0.8\left(1-\frac{s}{S_{o}}\right)}\;\;\;S_{o}\leq s \end{array}\right.,$$
(9)$$S_{o}=3\tau_{\max}\left[\frac{t_{f}}{G_{f}}+\frac{t_{a}}{G_{a}}+\frac{2h_{\mathrm{eff}}}{G_{c}}\right],$$
(10)$$\tau_{\max}=\frac{f_{t}}{1-\frac{\alpha }{\sqrt{\alpha^{2}+4}}},$$
$$\rho =\frac{A_{f}}{A_{c}},\;\lambda^{2}\;=\;\left[\frac{\Gamma G_{a}}{E_{f}A_{f}t_{a}}\left(1\;+\;\eta \rho \right)\right],\;\eta \;=\;\frac{E_{f}}{E_{c}},\;\alpha \;=\;\frac{\rho \Gamma_{f}}{\lambda A_{f}}$$
where $A_{c}$ is the effective area of cross section of the concrete (taken as to $6h_{\mathrm{eff}}b_{f}$ where normal stresses concentrate).

### 4.3. Modelling of Geometry

A detailed 3D brick element with 8-node used for simulation of the behaviour of the concrete with three DOF at each node used in publications [19,58] for the externally bonded FRP flexural strengthening is presented in Figure 8 for NSM FRP application. The steel rebar was simulated using 3-node line elements with three translational DOF at each node. The CFRP strips are represented using 4-node thin shell elements, with three translational degrees of freedom at each node.

Interface elements were used to connect the CFRP nodes and the concrete nodes. These elements were aligned in two perpendicular directions. The interface elements were used in both the longitudinal and vertical beam directions (Detail C in Figure 8). The element sizes of the concrete were selected to be a 12.5 mm cube to allow the finer meshing process at the interface of CFRP and concrete.

To capture the same failure modes that were observed experimentally (Figure 4 and Figure 5), the critical discrete crack approach was used to represent the geometric discontinuity arising from crack opening. These interface elements were aligned perpendicular to the two anticipated splitting planes as shown in Figure 8. From the experimental studies of this work and those available in the literature, it was concluded that the CFRP strip detaches from the surrounding concrete layer along with the level of the longitudinal steel reinforcement bars (Detail B, Figure 8). Thus, the predicted splitting planes (i.e., planes on which splitting cracks are initiated and propagate) were the two surfaces shown in Figure 8 (the plane between the steel reinforcement bars and CFRP strips and the plane for the flexural crack at the end of CFRP laminate). Figure 10 shows the typical finite element mesh of a strengthened modified beam.

## 5. Numerical Results and Discussion

The numerical results of stress $\tau_{\max}$, slip S_0_ and failure loads corresponding to all CFRP-strengthened beams are listed in Table 5.

### 5.1. Ultimate Loads and Failure Mode

The average ratio of numerical and experimental maximum load and its standard deviation were 1.02 and 0.13, respectively, indicating a good agreement with the test results. In all modelled beams, the numerical analysis simulated the concrete cover splitting failure mode. The comparison of the numerical to the experimental ultimate loads shown in Figure 11 reveals an excellent capability of the numerical and experimental results.

For the modified beam specimens, two failure modes, namely CFRP debonding and flexural failure, were expected to govern the ultimate capacity of the NSM CFRP-strengthened modified beams. In the finite element simulations, the CFRP debonding load was determined when the slip values in the horizontal interface elements connecting the CFRP strips and concrete nodes reached the maximum value (interface element group 2, Detail C in Figure 8). The flexural failure was controlled numerically by the opening of the major discrete crack at the end of the CFRP strip (interface element group 1, Detail B in Figure 8).

The effect of the NSM bond length on the ultimate load of the beams reinforced with lower steel reinforcement (Type A, reinforced with bars of 8 mm diameter) is presented in Figure 12. The solid curve presents the ultimate capacity of a tested specimen referring to the failure model controlled by a major flexure initiated at the end of the CFRP laminate. The dashed curve represents the debonding failure load of the simulated beam (due to bond failure). For design purposes, the maximum load of the beam indicates lower values of the flexural and debonding loads (solid and dashed curve, respectively). This indicates that for beams strengthened with NSM CFRP strips with bond lengths longer than 270 mm, the failure mode was controlled by debonding of the CFRP strips without any significant enhancement in the ultimate capacity over the load level corresponding to a bond length of 270 mm.

With an increase in the NSM bond length, both the debonding and flexural loads increased. For the beam strengthened with a lower bond length than 270 mm, debonding of the NSM strips from the concrete occurred. However, because there was no significant difference between the debonding load (dashed curve, Figure 12) and the flexural capacity (solid curve, Figure 12) of the beam with a bond length less than 270 mm, the failure mode could be a combination of the CFRP debonding and the flexural failure (observed in the form of formation of a plastic hinge in the vicinity of the major flexural crack at the end of the CFRP strip). For a relatively longer bond length (>270 mm), which was not experimentally confirmed, debonding of the CFRP strip could be controlled by the flexure.

### 5.2. Axial Strains in NSM CFRP Strips

The comparison of experimental and numerical load–strain relationships in the CFRP NSM strips are represented in Figure 13 for five selected specimens. It shows very good compatibility of numerical and test results.

The next comparison between numerical and test results of the compressive concrete strain shown in Figure 14 (based on LVDTs on the lateral side of the beams, Figure 3) confirms a good agreement between numerical and experimental results along the beam span. Small discrepancies observed between the experimental and numerical results around the failure load for the specimens NIISB/40/120 (Figure 14) is due to the fact that it closed to the failure load the strain gauges that failed during the concrete cover splitting. The comparison of test and numerical results presented in Figure 14 show that the finite element model successfully predicts the strain values in the concrete, particularly at the mid-span. Significant discrepancies occurred between both experimental and numerical strain readings around the end of the curve (values at the length of 900 and 1200 mm in Figure 14). The reason for these differences comes from the indirect measuring of the concrete strains in the experimental program using LVDTs that, due to a high length of the measured base (200 mm), did not reflect the real deformability of concrete.

Good predictive capabilities between numerical and experimental results encourage to use of this model for the prediction of the experimental results, which cannot be tested. Based on this assumption, the interfacial slip between the bonded NSM CFRP strips and concrete can be used to better understand the behaviour of the interface between the CFRP and concrete. For all tested specimens, the interfacial slip profiles had almost the same trend. The interfacial slip curves along the bonded laminate for a load level of 25% of the ultimate load and at the failure load for selected specimen NIISB/40/120 are presented in Figure 15. With the load increase up to the cracking load, the interfacial slip increases gradually along the bond length. It was noted that the increase in the slip value was associated with an abrupt increase in the slip at the CFRP strip end. At the cracking load level (25% of the failure load), the interfacial slip distribution was like that derived from pull-out tests of CFRP sheets externally bonded to concrete. At the failure load, a fluctuating slip was observed at the particular location of the first flexural crack.

The maximum interfacial slip value for all tested specimens is significantly lower than the value $s_{0}$ (Table 5). This indicates that the bonded joint did not reach its maximum capacity and the maximum bond stresses are still less than the bond strength of the interface ($\tau_{\max}$ in Table 5). It was confirmed in the test results [48] that observed failure modes of all tested specimens were caused by the flexural failure and all the CFRP–concrete interfaces did not reach their maximum capacity.

### 5.3. Numerical Aspects of the Simulations

Various aspects concerning the accuracy of the presented numerical simulations based on the interfacial fracture energy of the bond-slip model used to constitute the interface elements between the CFRP and concrete nodes in two perpendicular directions (Detail C, Figure 8) and the interfacial strength of the discrete crack approach (Detail B, Figure 8) are investigated in this study. The interfacial fracture energy is defined as the energy per unit bond area for complete debonding.

Two attempts were considered to investigate the effect of the interfacial bond-slip model for CFRP to concrete on the predicted results. The first attempt addressed the effect of bond-slip characteristics of the interface elements aligned in the horizontal direction between the CFRP nodes and steel nodes, while the second attempt considered interface elements in the vertical direction. It was observed that the interfacial stiffness and bond strength had a minimal influence on the overall structural stiffness when they were used to constitute the horizontal interface elements. However, they did have a significant effect when they were defined by the interfacial behaviour of interface elements in the vertical direction. The best example is the final element simulation of the Specimen NISA/20/85, decreasing the interfacial fracture energy from 2.45 N/mm to 0.65 N/mm, which resulted in a reduction in the ultimate capacity by 22.0%. It should be mentioned that by using a full bond behaviour for the interface between the CFRP strips and concrete nodes in the vertical direction, a significantly higher calculated strength was achieved (equal to 85% for the specimen NISA/20/85).

Moreover, the numerical analysis is very sensitive to the bond stiffness of the vertical interface elements connecting the CFRP element nodes and the concrete nodes. The influence of the concrete fracture energy used to constitute the bond behaviour of the major discrete crack interface element was caused by changing the concrete tensile strength for these interface elements from 1.6 to 3.2 MPa (beam NISA/20/85).

For the beams with a small bond length and failure due to debonding of the strip, it was found that the material properties of the discrete crack model did not affect the prediction. However, for a relatively long bond length (higher than 270 mm, Figure 12), the concrete fracture energy has a significant effect on the predicted failure load. It was observed that, in the numerical simulations, using the concrete tensile strength value to define the bond strength of the discrete crack interface element connecting the steel and concrete nodes underestimated the predicted values. The bond strength values that gave good predictions were around 1.5–2 times the concrete tensile strength.

## 6. Conclusions

The paper presents the selected test results published in [48] and a discussion of investigated parameters and their interactions on the CFRP to concrete bond behaviour: the beam span, beam depth, steel reinforcement ratio, number of CFRP strips, bond length and compressive concrete strength. Moreover, an advanced numerical analysis of each tested member was performed and compared with the test results. Based on the above analysis, the following conclusions may be drawn:–The test results confirmed that failure of NSM CFRP-strengthened specimens with continuous steel reinforcement was caused by the steel yielding and the flexural failure modes, rather than bond failure at the CFRP–concrete interface. The mode of failure was distinguished by the initiation of a major flexure crack at the end of the CFRP strip due to yielding of the internal reinforcement at this location, followed by complete debonding of the CFRP strips from the surrounding concrete beam. However, the slippage between the CFRP strips and concrete was observed in the beams with cut bars at the mid-span (specimens with index “s”).–The enhancement in the ultimate load-carrying capacity of the tested specimens was in the range of 6% to 24% of the corresponding control specimens for the short beams (NIS) with lower reinforcement (2 bars of 8 mm diameter) and in the range of 20% to 73% of the corresponding control specimens for the long beams (NIL) depending on the bond length of the NSM CFRP strips and concrete compressive strength.–It was unexpected that short strengthened beams with the higher reinforcement ratio (NIIS) failed under lower loading than the reference beam. It confirms that the reason for this failure mode is much stiffer behaviour of the beams reinforced with much higher internal reinforcement that caused such brittle behaviour of the strengthened beams after CFRP debonding.–The cutting of the longitudinal rebars delayed the debonding of the NSM laminates, thus significantly increasing the CFRP strains at failure. However, the beams with cut steel reinforcement at the mid-span indicated a 57% higher ultimate strain in the strengthening NSM CFRP strip (for the beams NILA/40/120 and NILA/40/120s). However, the ultimate load for the beam NILA/40/120s with cut reinforcement was 82% lower than this one with longitudinal reinforcement.–A slightly different situation was observed in the short beams with the higher reinforcement (NILB/40/120 and NILB/40/120s). The cutting of the longitudinal bars delayed the debonding of the NSM laminates and significantly increased the CFRP strains’ failure by 76%. However, the ultimate load for the beam with cut reinforcement NILB/40/120s was unexpectedly lower by 452% than this one with longitudinal reinforcement.–The experimental results showed that the steel reinforcement ratio is the most dominant parameter affecting the bond behaviour between the CFRP strips and concrete. Furthermore, it was observed that the final failure modes are mainly controlled by the effect of the internal steel bars (longitudinal and cut).–The proposed finite element model represents an advanced numerical tool based on micromechanics materials. The prediction of the ultimate load-carrying capacities was based on the major discrete crack approach to represent the flexural failure mode. The finite element analysis revealed a good efficiency of the predicted ultimate loads compared to the experimental ones with an average ratio of numerical and experimental maximum load and its standard deviation equal to 1.02 and 0.13, respectively.–The interfacial stiffness and the bond strength had a small influence on the overall structural stiffness when they were used to constitute the horizontal interface elements. However, they did have a significant effect when they were defined by the interfacial behaviour of interface elements in the vertical direction.–For the beams with a small bond length and failure due to strip debonding, the material properties of the discrete crack model did not affect the numerical predictions. For the specimens with a bond length higher than 270 mm, the concrete fracture energy significantly affected the ultimate load. The numerical simulations revealed that using the concrete tensile strength to define the concrete–steel bond strength with the discrete crack interface element underestimates the predicted values. The bond strength that gave quite good predictions was between 1.5–2.0 times the concrete tensile strength.

## Figures and Tables

**Figure 1 materials-14-04362-f001:**
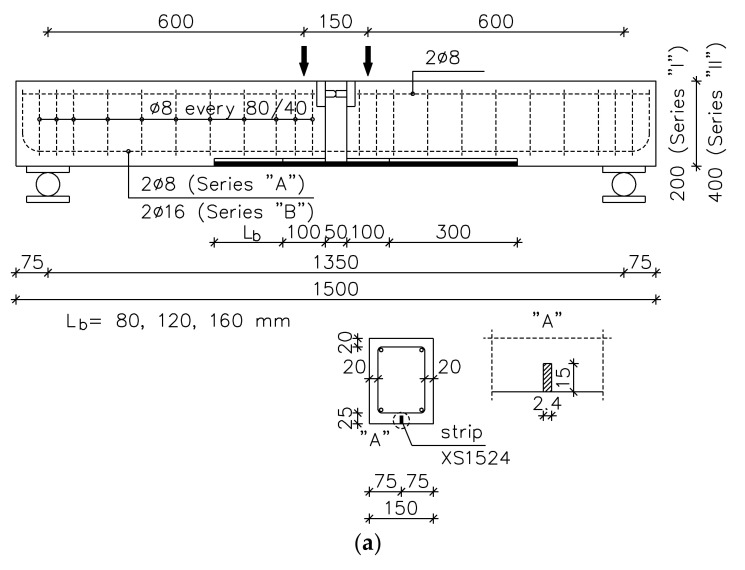
Dimensions (in mm), steel reinforcement and modes of strengthening: (**a**) short—S; (**b**) long beam specimens—L [48].

**Figure 2 materials-14-04362-f002:**
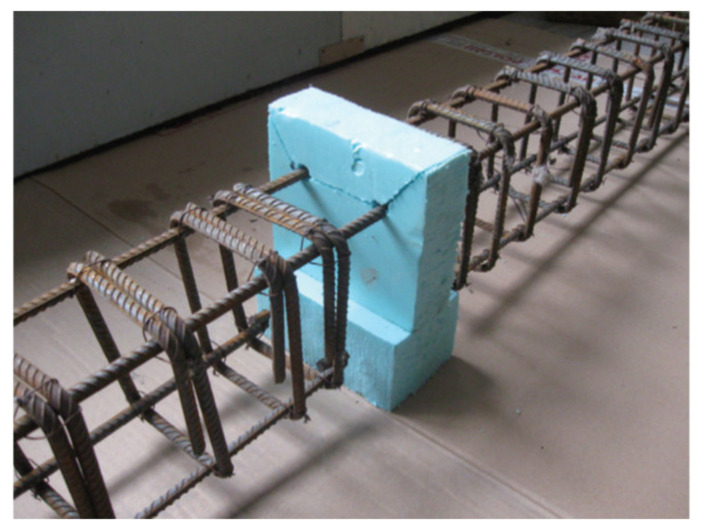
Midspan part of the steel reinforcement skeleton of Series I beams with the foam polystyrene [48].

**Figure 3 materials-14-04362-f003:**
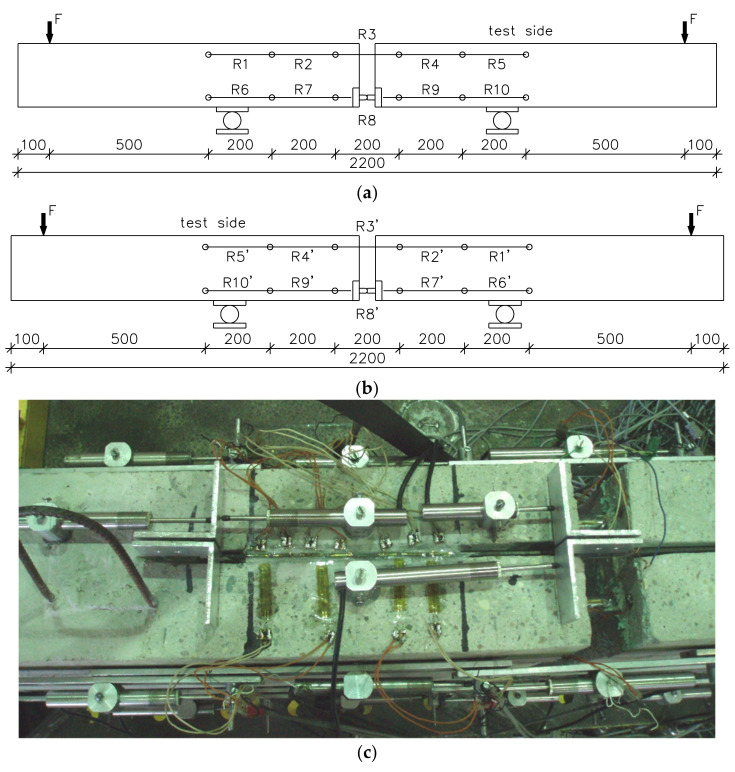
Location of strain measurements (dimensions in mm): (**a**) axial strains in compression and tension on the test site and (**b**) opposite site; (**c**) strain gauges bonded to the concrete and CFRP strip with LVDTs at the end-bond slip measurements on loaded and free CFRP strip [48].

**Figure 4 materials-14-04362-f004:**
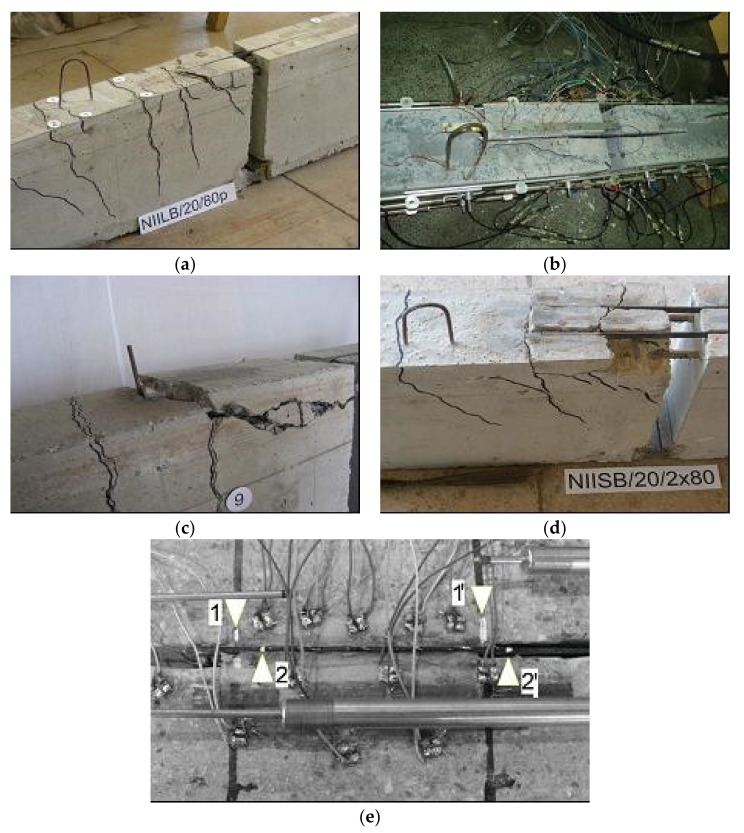
Different failure modes of specimens: (**a**) NIILB/20/80; (**b**) NISA/20/160; (**c**) NISA/20/80; (**d**) NIISB/20/2 × 80; (**e**) NILB/40/120s [48].

**Figure 5 materials-14-04362-f005:**
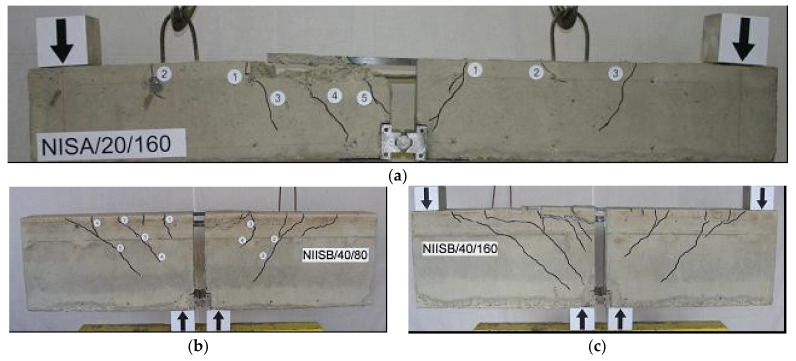
Crack patterns of the selected specimens: (**a**) beam NISA/20/160, (**b**) beam NIISB/40/80, (**c**) beam NIISB/40/160 [48].

**Figure 6 materials-14-04362-f006:**
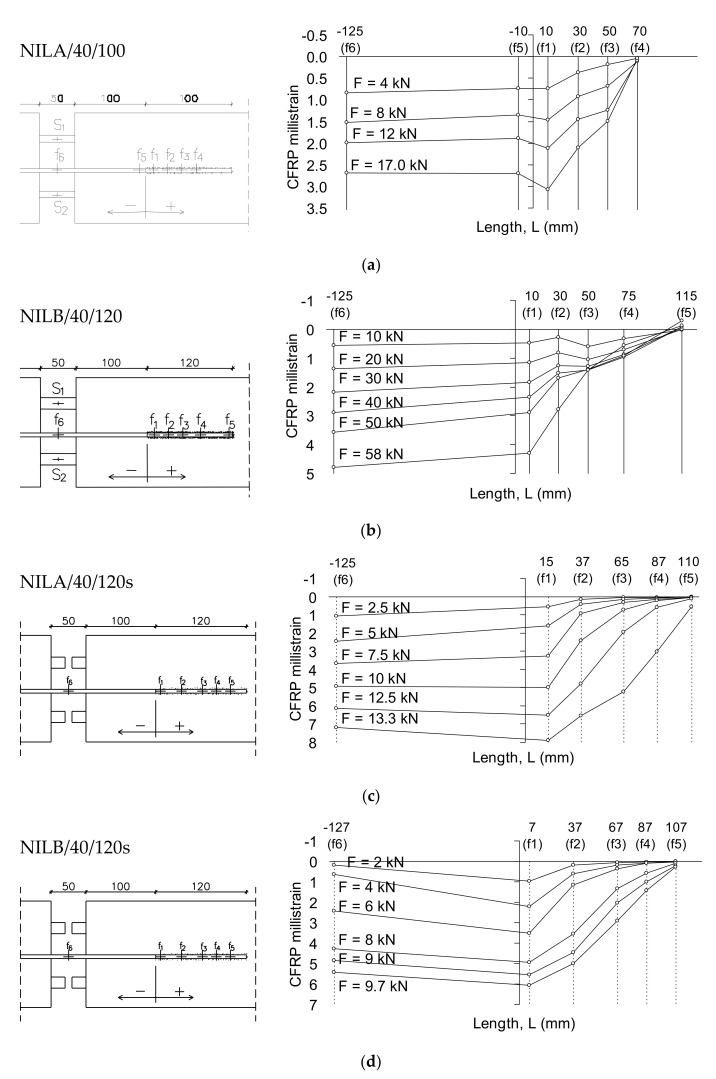
Experimental CFRP strain profiles for beams: (**a**) NILA/40/100, (**b**) NILB/40/120, (**c**) NILA/40/120s and (**d**) NILB/40/120s [48].

**Figure 7 materials-14-04362-f007:**
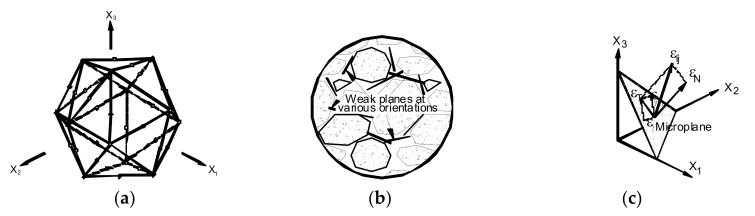
Representative material element in microplane model: (**a**) spatial distribution of microplanes, (**b**) weak planes at level, (**c**) decomposition of strain microstructure tensor on a microplane [71]).

**Figure 8 materials-14-04362-f008:**
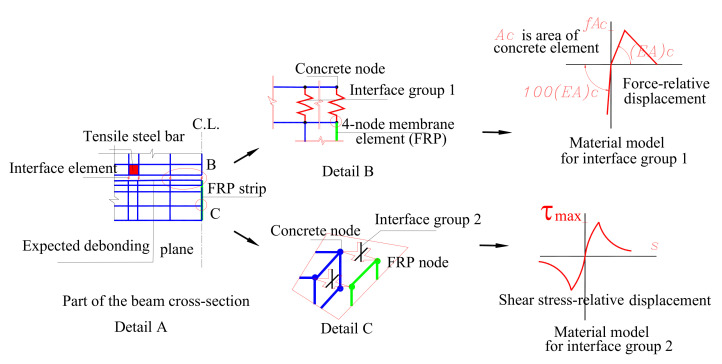
Typical finite element model for NSM specimens.

**Figure 9 materials-14-04362-f009:**
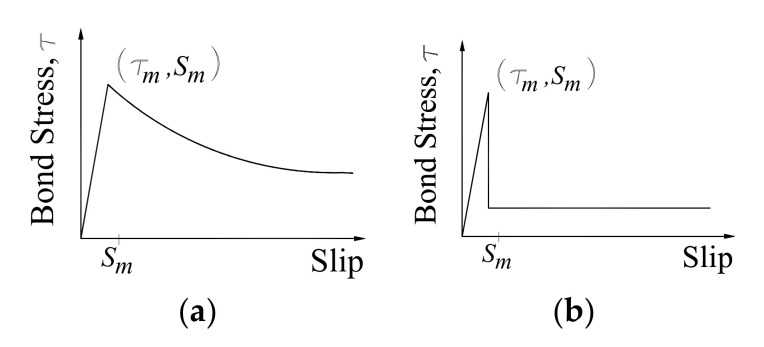
Bond-slip characteristics of NSM FRP to concrete: (**a**) failure inside concrete; (**b**) failure at bar-adhesive interface [38].

**Figure 10 materials-14-04362-f010:**
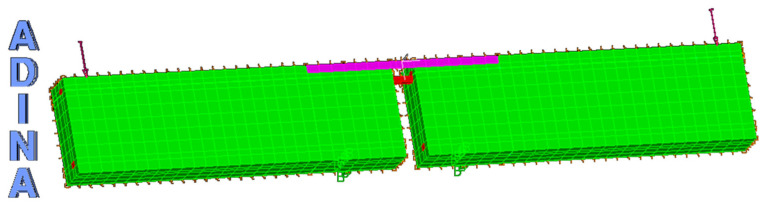
Typical finite element mesh.

**Figure 11 materials-14-04362-f011:**
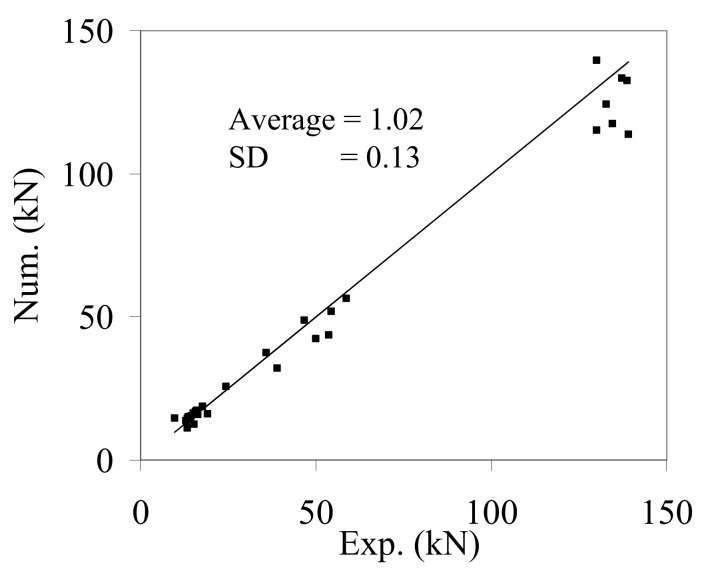
Comparison between the experimental and numerical maximum load.

**Figure 12 materials-14-04362-f012:**
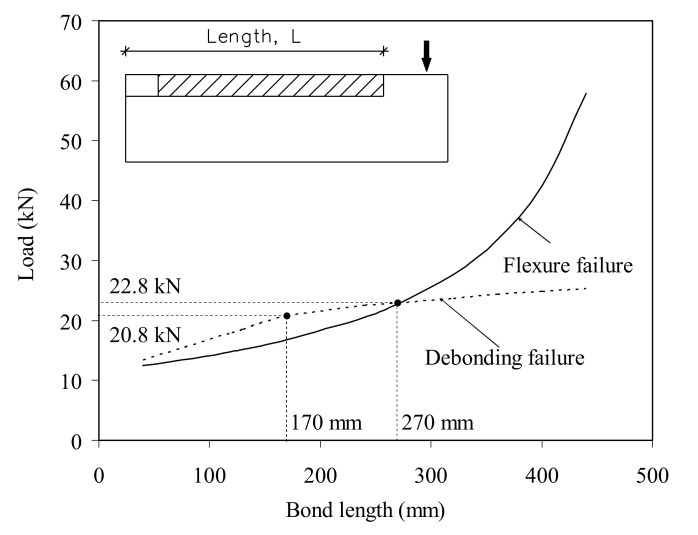
Effect of the bond length on the predicted ultimate load for the beams of Type A.

**Figure 13 materials-14-04362-f013:**
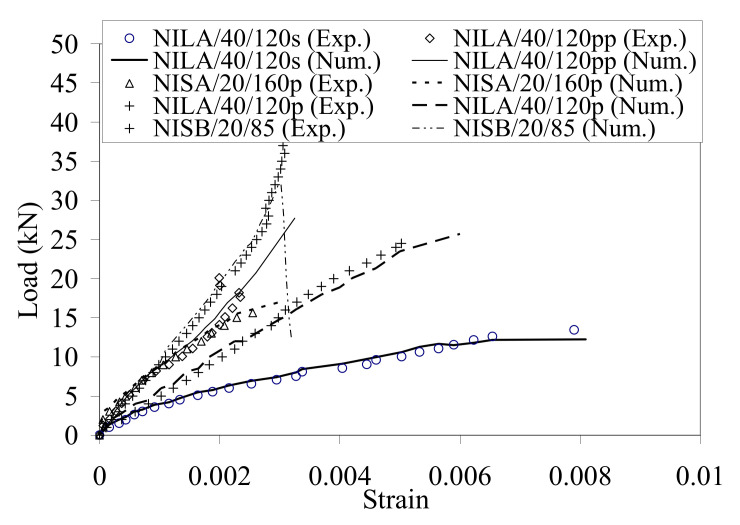
Load–strain profiles in the NSM strips for selected five specimens.

**Figure 14 materials-14-04362-f014:**
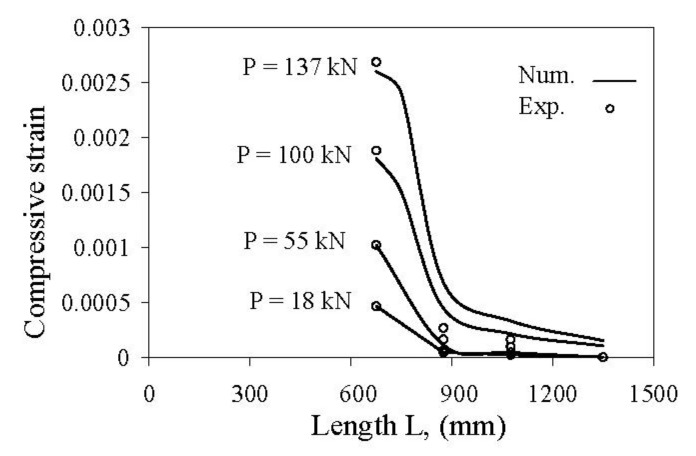
Compressive concrete strains curve along the beam span for beam NIISB/40/120.

**Figure 15 materials-14-04362-f015:**
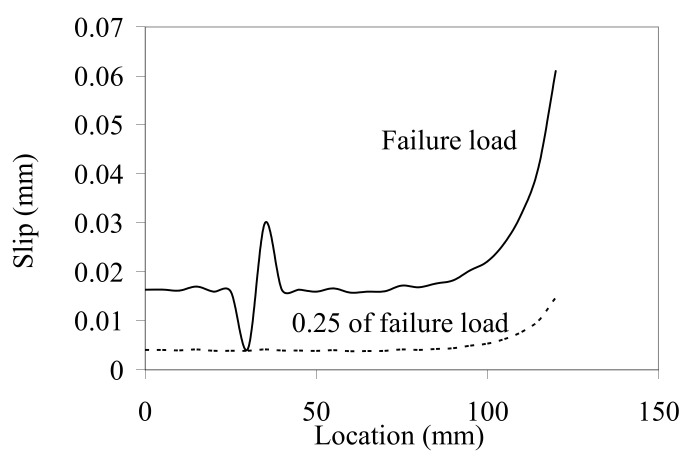
Interfacial slip distributions for specimen NIISB/40/120.

**Table 1 materials-14-04362-t001:** Composition of the concrete mixture [48].

Component	Amount (kg/m^3^)	
	$f_{c}^{\prime }=20\mathrm{MPa}$	$f_{c}^{\prime }=40\mathrm{MPa}$
Coarse aggregate	625	812
Fine aggregate	625	541
Cement	300	400
Water/Cement ratio	0.6	0.43

**Table 2 materials-14-04362-t002:** Concrete properties and test results [48].

No	Series ID	Steel Reinforcement Ratio	Beam ID	$f_{c}^{\prime }$ MPa	$f_{t}$ MPa	$L_{b}$ mm	$F_{sy}$ kN	$F_{u}$ kN	$\frac{F_{u}}{F_{u0}}$	$\varepsilon_{su}$	$\varepsilon_{fb}$
1	NIS	2#8	NISA/20 *	24.50	–	–	–	14.16	–	–	–
2			NISA/20/85	22.30	1.6	85		13.27	0.94	0	0.00208
3			NISA/20/120	21.30	2.0	120	13.40	15.21	1.07	0.00287	0.00260
4			NISA/20/130	23.00	1.6	130	14.98	14.98	1.06	0.00255	0.00193
5			NISA/20/160	21.30	2.0	160	15.20	15.63	1.10	0.00286	0.00258
6			NISA/30	32.50	–	30	10.70	12.89	–	0.00273	–
7			NISA/30/80	32.50	–	80	15.60	15.96	1.24	0.00284	0.00264
8			NISA/30/120	32.50	–	120	11.01	14.40	1.12	0.00276	0.00316
9		2#16	NISB/20 *	19.84	2.1	–	32.20	49.91	–	0.00268	–
10			NISB/20/85	19.84	2.1	85	32.21	38.92	0.78	0.00268	0.00293
11			NISB/20/130	19.84	2.1	130	34.76	35.79	0.72	0.00285	0.00264
12	NIIS	2#16	NIISB/40/80	41.58	3.8	80	110.00	130.00	–	0.00245	0.00337
13			NIISB/40/2 × 80	41.19	3.8	80	129.9	129.99	–	0.00234	0.00198 (0.00246)
14			NIISB/40/120	41.19	4.4	120	116.18	137.20	–	0.00265	0.00351
15			NIISB/40/160	41.19	4.4	160	119.47	138.65	–	0.00286	0.00416
16	NIL	2#8	NILA/40 *	38.30	3.2	–	10.98	14.01	–	0.00275	–
17			NILA/40/100	41.75	3.8	100	12.00	16.29	1.20	0.00286	0.00272
18			NILA/40/120	38.50	3.4	120	19.00	24.30	1.73	0.0029	0.00502
19			NILA/40/160	38.50	3.4	160	16.28	17.64	1.26	0.00273	0.00282
20			NILA/40/120s	45.00	3.9	120	–	13.33	–	–	0.00791
21			NILA/50	47.50	2.8	–	11.24	13.60	–	0.00244	–
22			NILA/50/2 × 80	47.50	3.3	80	18.07	19.05	1.40	0.00281	0.00263 (0.00240)
23		2#16	NILB/40 *	38.30	3.3	–	35.00	46.61	–	0.00424	–
24			NILB/40/90	37.67	3.1	90	40.00	54.33	1.17	0.00281	0.00391
25			NILB/40/120	37.67	3.1	120	40.04	53.60	1.15	0.00275	0.00345
26			NILB/40/130	43.70	–	130	50.75	58.60	1.26	0.00311	0.00358
27			NILB/40/120s	43.70	–	120	–	9.70	–	–	0.00606
28	NIIL	2#16	NIILB/40/80	34.32	4.0	80	115.00	139.07	–	0.00273	0.00464
29			NIILB/40/2 × 80	34.32	3.3	80	130.31	134.48	–	0.00249	0.00274 (0.00325)
30			NIILB/40/120	38.80	3.3	120	132.74	132.74	–	0.00305	0.00358

* Control beam; $\varepsilon_{fb}$: axial strain of the CFRP at the ultimate load of the specimen; $\varepsilon_{su}$: axial strain of the internal steel bars at the ultimate load; $F_{u}$, $F_{u0}$: ultimate loads of the strengthened and unstrengthened beams, respectively; $F_{sy}$: yield load; s: beam with cut steel bars.

**Table 3 materials-14-04362-t003:** Mechanical characteristics of steel reinforcement [48].

Diameter (mm)	$E_{s}(\mathrm{GPa})$	$f_{su}(\mathrm{MPa})$	$f_{ys}(\mathrm{MPa})$
8	207	637	543
16	209	636	542

**Table 4 materials-14-04362-t004:** Mechanical characteristics of CFRP strips [48].

Type	Width	Thickness	$f_{fu}(\mathrm{MPa})$	$E_{f}(\mathrm{GPa})$	$\varepsilon_{fu}$
XS1.524	15.10	2.41	36.39	169.4	0.0112

**Table 5 materials-14-04362-t005:** Bond-slip properties and predicted load capacities.

No	Beam ID	$\tau_{max}$ (MPa)	$S_{0}$ (mm)	$F_{u,num}$ (kN)	$F_{u,num}/F_{u,exp}$
1	NISA/20 *	-	-	14.00	1.01
2	NISA/20/85	6.44	0.0663	11.18	1.19
3	NISA/20/120	6.14	0.0651	12.48	1.22
4	NISA/20/130	6.44	0.0663	16.35	0.92
5	NISA/20/160	6.14	0.0651	16.96	0.92
6	NISA/30	-	-	13.75	0.94
7	NISA/30/80	7.96	0.0721	17.34	0.92
8	NISA/30/120	7.96	0.0721	15.45	0.93
9	NISB/20 *	-	-	42.41	1.18
10	NISB/20/85	5.87	0.0640	32.10	1.21
11	NISB/20/130	5.87	0.0640	37.50	0.95
12	NIISB/40/80	9.26	0.0767	139.70	0.93
13	NIISB/40/2 × 80	9.21	0.0766	115.26	1.13
14	NIISB/40/120	9.21	0.0766	133.43	1.03
15	NIISB/40/160	9.21	0.0766	132.60	1.05
16	NILA/40 *	-	-	15.04	0.93
17	NILA/40/90	9.28	0.0768	15.89	1.03
18	NILA/40/120	8.84	0.0752	25.71	0.95
19	NILA/40/130	8.84	0.0752	18.75	0.94
20	NILA/40/120s	9.72	0.0783	14.59	0.91
21	NILA/50	-	-	15.11	0.90
22	NILA/50/2 × 80	10.04	0.0795	16.14	1.18
23	NILB/40 *	-	-	48.87	0.95
24	NILB/40/90	8.72	0.0748	51.95	1.05
25	NILB/40/120	8.72	0.0748	43.69	1.23
26	NILB/40/130	9.55	0.0777	56.48	1.04
27	NILB/40/120s	9.55	0.0777	14.61	0.66
28	NIILB/40/80	8.24	0.0731	113.80	1.22
29	NIILB/40/2 × 80	8.24	0.0731	117.54	1.14
30	NIILB/40/120	8.88	0.0754	124.35	1.07

* Control beam.

## Data Availability

The data presented in this study are available on request from the corresponding author.

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
