# Peer review of "Bond Behaviour of Near-Surface Mounted Strips in RC Beams—Experimental Investigation and Numerical Simulations"

_materials, 2021, doi:10.3390/ma14164362_

Round 1

Reviewer 1 Report

The manuscript is well written and addresses bond behavior for near-surface mounting (NSM) technique using CFRP strips for rehabilitation of reinforced concrete beams. The experimental results augmented by the numerical simulations are well explained and no further additions are required for the defined scope of the paper. Following are specific comments/observations for further improvements.

  1. Figure 6 and line 335-346. This section should be written again as there is no coherence between the specimen designations, Figure 6(a-d) and the numerical values from Table 2, resulting in unclear discussion. Figure 5d is cited that is actually Figure 6d.
  2. Different referencing styles have been used in section 1-3 and the later sections. This should be consistent with respect to the format guidelines by the Journal.
  3. Line 364. Correct start of the sentence ‘This the’
  4. Figure 7. Detail A, B and C is not clear with respect to the test specimens. This diagram needs improvement and more detailing as which part of cross section is under discussion. Definition of material models is fine.
  5. Check cited reference Abdel Baky et al. (2009), published in 2007 numbered as [19]?
  6. Line 409. Expression for lamba should be moved to the end after alpha.
  7. Line 412-413. Gf is shear modulus in transverse or longitudinal direction of FRP? Please check. An and what is Ga, shear modulus of adhesive?
  8. Line 424-426. The direction of CFRP and interface elements as defined is not clear for the reader. The sentence needs clarification / rephrasing.
  9. Line 481-485. In Figure 10, dashed line represents ‘debonding’ and solid line represents ‘flexural failure’, however, opposite is mentioned in the text.
  10. Figure 11. It represents load strain profiles for 5 specimens, However, line 492 reads 2 specimens. Check for consistency.
  11. Figure 11. Legend is not in good order, difficult for the reader to correlate the results. Represent marks for experiments on left and corresponding numerical on right.
  12. Figure 12. How the compressive strains along the beam span were measured experimentally? The direction of the strain gauges in figure 3 is transverse that doesn’t allow measurement in longitudinal direction.
  13. Line 507. LDVDs or LVDTs?
  14. Figure 13. This is the numerical output or experimental? Is this possible to compare in this diagram? The jump in the slip makes no sense. Please explain.
  15. References are named as Acknowledgements.

Author Response

The authors would like to thank the Reviewer for the thorough examination of the manuscript and valuable comments, which will undoubtedly enhance the manuscript’s quality. The authors have tried to attend to the comments and suggestions of the Reviewer in the following, revised version of the manuscript. All the comments made by the Reviewer have been addressed individually and presented in the following order: the Reviewer’s comment/question is presented first (in italic), then the authors’ answer is given next. Corrections to grammatical, spelling, stylistic errors and other minor imperfections, suggested by the reviewers, were appended in the revised manuscript.

Reviewer 2 Report

Overall the research and presentation is good. I suggest removing the Appendix and just keeping the reference to the two cited works.

Author Response

(The authors gave the same response as above.)

Reviewer 3 Report

The bond behavior of CFRP is an important topic and should be designed properly to ensure flexural performance after the repair. And Overall, the presentation and contents are fairly well done to show the experimental program and results/analytical results. However, the literature review should be enhanced by reviewing recent articles. General one idea is grouping the most recent literature. For example, the one sentence represents all the recent findings with referring to reference 21-30.  The literature review section should be improved to strengthening the needs of this study. 

I would suggest removing the sentence in the introduction (lines 94-96). The introduction doesn't need the results such as "good agreement" between analytical results and test results. This is not the purpose of this study. The reviewer highly recommends rewriting this sentence to strengthen the needs of this study and how this study is different from previous research except for the words "the precisely described. "

In the introduction:

Line 43-43: "Add more elaboration on the recent research." Add more literature review on recent papers and justify the need of this research. Other literatures are outdated to reflect the recent research.

 Line 93-96. "Rewrite the purpose of the research" The objective is not presented.

Author Response

(The authors gave the same response as above.)

Reviewer 4 Report

The present paper investigates the bond mechanism between carbon fibre reinforced polymer (CFRP) laminates, concrete and steel in near-surface mounted (NSM) CFRP-strengthened reinforced concrete (RC) beams. The investigation of five parameters was investigated. Furthermore, a FE model has been developed. Figures and tables are clear. The english is good.
1) In the first part of the introduction section, I recommend considering the following state of the art paper about FRP system and other strengthening systems in which adhesion play a crucial role:
 -) 10.3390/cryst11030265
 -) 10.1007/978-3-030-41057-5_31
2) I suggest putting the paper’s outline at the end of the introduction section to improve the paper’s readability.
3) Can you please provide more information about the mix design adopted to obtain 20 and 40 f’c.
4) Is f’c a characteristic or a medium value?
5) In my view, it is not clear if the space generated by the foam has been covered by using concrete or not.
6) I suggest putting more data about the FE model. i.e. shape function is adopted etc.
7) The conclusion section may be improved.

Author Response

(The authors gave the same response as above.)

Round 2

Reviewer 4 Report

The manuscript can be published in the present form